# Goalkeepers Live Longer than Field Players: A Retrospective Cohort Analysis Based on World-Class Football Players

**DOI:** 10.3390/ijerph17176297

**Published:** 2020-08-29

**Authors:** Witold Śmigielski, Robert Gajda, Łukasz Małek, Wojciech Drygas

**Affiliations:** 1Department of Demography, University of Lodz, 90-131 Lodz, Poland; 2Center for Sports Cardiology at the Gajda-Med Medical Center in Pułtusk, 06-100 Pultusk, Poland; gajda@gajdamed.pl; 3Department of Epidemiology, Cardiovascular Disease Prevention, and Health Promotion, Cardinal Stefan Wyszynski National Institute of Cardiology, 04-628 Warsaw, Poland; lmalek@ikard.pl (Ł.M.); wdrygas@ikard.pl (W.D.); 4Department of Preventive and Social Medicine, Medical University of Lodz, 90-131 Lodz, Poland

**Keywords:** football goalkeepers, football field players, football, life expectancy, longevity, playing position

## Abstract

The purpose of this article is to study whether the position occupied by footballers on the pitch influences their life duration. It is known that various types of sporting activity (endurance, resistance, or mixed) may influence lifespan in different ways. However, there is a paucity of data regarding the role of different positions played in team sports such as football. Our research was based on elite international football players born before 1923 who took part in the first three football World Cups (*n* = 443) or played in the 1946/1947 season in the leading clubs of the main European leagues (*n* = 280). Goalkeepers were characterized by a 5–8-year longer life duration compared to their colleagues playing in other positions (World Cup: 82.0 ± 7.0 vs. 74.0 ± 8.0, *p* = 0.0047; European leagues: 83.0 ± 7.5 vs. 78.0 ± 8.0, *p* = 0.0023), with an absence of differences between defenders, midfielders, and forwards. Moreover, in both of the analyzed subgroups, the rate of survival until the 85th birthday was significantly higher among goalkeepers than among field players (*p* = 0.0102 and *p* = 0.0048, for both studied groups, respectively).

## 1. Introduction

Regular physical activity is one of the important factors influencing human health and longevity [1,2,3,4,5]. The influence of competitive sports on human longevity remains controversial. Although large clinical studies and meta-analyses in elite athletes have demonstrated a longer lifespan in that group, some authors have emphasized the negative aspects of professional sports, which are related to excessive training volume or intensity, risk of serious injuries, or illegal pharmacological support [6]. Great pressure from fans, and stress related to feelings of responsibility for the result of important games are among the negative factors that can influence the life expectancy of athletes. Another factor, which is not perfectly understood, is the lifestyle of elite athletes after ending their professional careers [7,8]. Alcohol and drug abuse as well as sexual promiscuity may sometimes accompany professional sporting careers, at least in some countries and sports disciplines.

Football is a game played by more than 120 million people and watched by approximately 3.5 billion fans worldwide. Despite the fact that football is extremely popular globally, there is a paucity of data analyzing how competitive football playing influences life duration [9,10,11]. Moreover, the available data rarely address the issue of position occupied by football players on the pitch [12]. Football is a complex game with an inconsistent level of intensity of physical effort during the game or training. Different physical abilities and morphological and physiological characteristics are required for different playing positions. Field players should possess high power and endurance, whereas goalkeepers need to have high explosive capabilities. Top-class goalkeepers have different anthropometric and functional features; similarly, their mode of training, exercise expenditure, and the results of many specific functional tests are different from those of outfield players. These characteristics can potentially have an impact on their life expectancy [13].

Therefore, the aim of this study was to analyze whether the lifespan of football players depends on the position occupied on the pitch.

## 2. Materials and Methods

### 2.1. Study Group

We analyzed the lifespan of football players born before 1922. The players took part in the first three World Cup tournaments (Uruguay 1930, Italy 1934, and France 1938) and/or played in the 1946/1947 season in the leading clubs of the main European leagues (English, Spanish, Italian, German, French, Scottish, and Swedish, i.e., Manchester United, Liverpool, FC Barcelona, Real Madrid, Juventus Turin, Inter Milan, Borussia Dortmund, Bayern Munich, AS Saint-Etienne, Hibernian FC, or Malmo FF). Restricting the research to only players born before 1923 allowed us to focus on complete observations of the duration of life. Any lack of information about the year of death is most likely to be associated with a lack of publicly available information, rather than with the fact that the player is still alive; if they were still alive, the athlete would be at least 98 years old.

### 2.2. Data Collection

The website www.worldfootball.net [14] was the main source of information; however, the collected data were confirmed and verified using other sources as well. In total, 921 competitors (545 players from the World Cup tournaments and 376 competitors from the European leagues) were analyzed. Full record data were obtained for 487 participants of the World Cup tournaments (89.4%) and 295 players of the abovementioned European leagues (78.9%). For methodological reasons, athletes who died during World War II (1939–1945) or shortly after it ended (1946–1950) were excluded from further analysis, which unfortunately occurred in 32 cases. Those playing for teams in which more than half of the competitors had missing reliable record data (which included the teams for Paraguay 1930, Cuba 1938, and Dutch East Indies 1938) were excluded. League players who appeared in the abovementioned World Cup finals were also omitted (to avoid duplication of data). The final sample size consisted of 723 players (443 participants of the World Cup tournaments and 280 European league players).

### 2.3. Statistical Analysis

Owing to a lack of normal distribution, which was tested using the Shapiro–Wilk test, the analysis of variance (ANOVA) Kruskal–Wallis test was used to compare the lifespan of players according to their field position (4 groups). Post hoc analysis for Kruskal–Wallis ANOVA was done by using a multiple comparison test (Dunn test). The Mann–Whitney test was used to compare differences in lifespan between goalkeepers and field players (analyzed together). Survival analysis was performed using Kaplan–Meier curves. Significance of frequencies of surviving to 85 years old was tested using a proportion test. All statistical calculations were performed using STATISTICA 12 PL (StatSoft, Krakow, Poland) software. The significance level was set at *p* < 0.05.

## 3. Results

The life duration of the analyzed football players is presented in Table 1.

The results of the first step of analysis (ANOVA Kruskal–Wallis test) indicated that the lifespan of football players differed significantly due to the players’ positions (*p* = 0.0285 and *p* = 0.0064, respectively). The post hoc analysis (Dunn test) for World Cup participants indicated that the lifespan of goalkeepers differed significantly in comparison to forwards and defenders (*p* = 0.0234 and *p* = 0.0475, respectively), but differences between the lifespans of goalkeepers and midfielders were insignificant (*p* = 0.1959). An analogous analysis for players of the European leagues indicated that the lifespan of goalkeepers differed significantly in comparison to defenders, and differences in lifespan between goalkeepers and midfielders or forwards were close to the significance level (*p* = 0.0919 and *p* = 0.1079, respectively). It is worth emphasizing that the lifespan of field players did not differ significantly among World Cup participants nor players of the European leagues (*p* = 0.7152 or higher). As such, one can analyze them together (i.e., defenders, midfielders, and forwards), thus revealing a difference in lifespan between goalkeepers and field players that was statistically significant (World Cup players: *p* = 0.0047; European Leagues: *p* = 0.0023; see Table 1). Moreover, in both subgroups, the rate of survival until the 85th birthday was significantly higher among goalkeepers than among other players with different positions (*p* = 0.0102 and *p* = 0.0048, respectively). Thus, it should be noted that the goalkeepers had a longer life duration by approximately 5–8 years compared to their colleagues playing in other positions (World Cup: 82.0 ± 7.0 vs. 74.0 ± 8.0 years, *p* = 0.0047; European leagues: 83.0 ± 7.5 vs. 78.0 ± 8.0 years, *p* = 0.0023), but no differences were found between defenders, midfielders, and forwards (World Cup: *p* = 0.5706; European leagues: *p* = 0.1971). It should be noted that the mean life expectancy of the male population in the second half of the 20th century was approximately 60 years worldwide and approximately 70 years in Europe. Therefore, the lifespan of football players (particularly goalkeepers) was markedly higher [15]. Survival probabilities when analyzing players with respect to their playing position are presented in Figure 1 and Figure 2.

Analysis of the survival curves showed significant differences between the goalkeepers and players in other positions on the pitch (*p* = 0.0285 and *p* = 0.064 for World Cup players and European league players, respectively) with a lack of significant differences between various field player categories (defender, midfielder, and forward). In particular, World Cup goalkeepers had a lower risk of death in the elderly stage of life (65–85 years), whereas goalkeepers playing in the leading European leagues had lower pre-elderly (60–65 years) and late elderly stage (85–90 years) mortality rates than field players.

## 4. Discussion

In an attempt to explain these rather surprising results, the authors would like to draw attention to several aspects that may be related to the longer life duration observed in goalkeepers.

### 4.1. Training Characteristics

Firstly, it should be noted that there are significant differences in the training programs and patterns of physical activity during games between goalkeepers and football players in the field.

On average, field players run approximately 10–12 km during one 90-min game, in comparison to the 4–6 km distance run by goalkeepers [16]. Moreover, goalkeepers spend approximately 75% of the match time at low movement intensities. Playing in the field is dynamic and involves short sprints at maximal effort. In addition to running, there is also twisting of the torso to change direction, heading, tackling, and holding the ball against defensive pressure [17]. Work intensity can approach the anaerobic threshold, which is defined as the highest exercise intensity, usually at 80–90% of the maximum heart rate [18]. Different forms of training and game efforts are required from goalkeepers. These include less running at a lower speed, and more gymnastics and resistance training in the gym [19]. Thus, the training of football goalkeepers is typically completed separately from outfield players, with a focus primarily on technical or explosive drills [16]. Routine goalkeeper training has an excess of accelerations/decelerations and a lack of running activities performed at high metabolic loads [20].

### 4.2. Sports Intensity and Survival

Available studies have demonstrated not only significantly lower aerobic capacity, but also a lower number of injuries among goalkeepers than among field players [21,22,23]. Similar results were also observed by Ziv and Lidor [24]. The amount and quality of physical activity performed by goalkeepers correspond well with the amount of physical activity recommended for optimal reduction of cardiovascular risk [25]. Many studies have demonstrated that a higher physical activity load does not bring additional health benefits. In fact, the relationship between physical activity and cardiovascular risk may have a U-shaped pattern, wherein sedentary lifestyle and excess physical training may be related to an increased risk of cardiovascular complications [25,26,27]. In a large retrospective study, Löllgen et al. demonstrated that light- and moderate-intensity activities were generally associated with a reduction in mortality, whereas training at high intensities was not required for the main prevention of all-cause mortality [28]. The physical activity pattern of field players is located on the right upslope of the U-curve and is certainly shifted rightward in comparison to goalkeepers. These observations were not confirmed, however, by Lemez and Beker in a large study analyzing the mortality and longevity of elite athletes. Considerable support was found for superior longevity outcomes for elite athletes, particularly those in endurance and mixed sports (e.g., football). However, the player’s position on the pitch was not taken into consideration in that study. The authors concluded that future research into the mechanisms that may affect mortality risk is important for better understanding of life expectancies in both eminent and non-eminent populations, but participation in elite sports is generally favorable to a longer lifespan [6]. In line with this, a large cohort study reported that cardiorespiratory fitness was inversely associated with long-term mortality, with no observed upper limit of benefit [29]. Another study indicated that the relationship between physical activity and cardiovascular risk should rather be described not as a U-shaped but as a J-shaped pattern, with no high limit of physical fitness that could be negatively correlated with risk of cardiovascular mortality [30].

The difference in the intensity and characteristics of physical activity may also influence the manifestation of congenital or acquired heart conditions. Abrupt and intensive start-stop bursts of exercise, which are more likely to occur in field players, were shown to serve as a trigger for arrhythmias and pose a higher risk of sudden cardiac death (SCD) [31,32].

SCD is defined as a natural death due to cardiac causes, characterized by an abrupt loss of consciousness within 1 h after the onset of symptoms [33]. SCD in athletes is most commonly caused by congenital or acquired cardiovascular disease. Studies performed on athletes in the United States found hypertrophic cardiomyopathy as the most common cause of SCD, followed by congenital coronary artery anomalies, myocarditis, and arrhythmogenic right ventricular cardiomyopathy. Ion channelopathies, such as long QT and Brugada syndrome, were also identified. SCD can also be induced by a traumatic blow to the chest (commotio cordis) [34,35]. In previous studies, it was not always possible to unequivocally determine the cause of death. Different terms were used for the cause of death, including SCD, cardiac arrest, or collapse during a match, often without information on the mechanism or disease responsible for the event.

Of note, there were no SCDs in the analyzed cohort. However, we were unable to establish whether any of the diseases described above might have influenced SCD off the pitch. In the analyzed group, five athletes died during their active sporting career (0.7%), but the causes of death remain unknown.

Our separate analysis of the causes of death of football players who died during their active sporting career between 1976 and 2019 showed that goalkeepers had a lower risk of death due to cardiovascular causes than field players, with the highest risk observed among defenders. As demonstrated in this analysis of 400 players, 14 deaths (3.7%) occurred during the game or training, or shortly after. Six of these deaths (1.5%) were due to cardiac arrest. Of note, none of the six deaths occurred among goalkeepers. Other non-cardiac causes of death during the game or training included body collision with trauma (mainly head injuries) or a fall of unidentified cause during physical activity [36].

### 4.3. Other Potential Factors

Another factor that might have been responsible for the observed fatality findings is head injuries. There are studies linking health risks associated with hitting the head. Repeated brain concussions caused by hitting the ball with the head, or during body collisions, often lead to chronic traumatic encephalopathy (CTE). CTE could increase the risk of premature deaths, mainly due to dementia [37,38,39,40,41]. With the abovementioned data in mind, it is worth noting that, unlike football players in field positions, goalkeepers are very rarely hit on the head by the ball.

Genetic factors should also be considered. Several gene polymorphisms have been described, although not unequivocally, as predisposing a person to an athletic career (*ACE*, *ACTN3*, *PPARA*, and *UCP2*). The two most studied gene variants are *ACE* and *ACTN3* polymorphisms [42]. In particular, a different prevalence of the angiotensin I-converting enzyme (*ACE*) insertion/deletion (ID) polymorphism in endurance and strength athletes was observed. The I allele, causing a lower *ACE* activity, was related to higher endurance performance. Conversely, the D allele was associated with strength and power secondary to increased concentrations of *ACE* and angiotensin II, which is a growth factor [43]. The second variant of potential relevance is the *ACTN3* gene, which encodes α-actining-3 protein, a structural protein of type-II muscle fibers. There are two allelic variants (R and X). Individuals with XX variants were found to have a lower percentage of these fibers and to be under-represented among elite athletes, particularly in sports where high endurance and strength are key factors of success [44]. Polygenic profiling of Russian football players, including player position as one of the factors, found that goalkeepers had an increased frequency of the *ACE* D allele, whereas attackers exhibited a higher frequency of the *ACTN3* R allele [42]. These differences may potentially impact the duration of life of these athletes. One meta-analysis has demonstrated that the *ACE* D allele has some selective advantages that contribute to longevity in the majority of European populations [45]. Another study on rugby players found that the *ACTN3* X allele was related to higher high-density lipoprotein levels with known cardioprotective effects [46]. On the other hand, other studies showed that carriers of the *ACE* D polymorphism had a higher risk of myocardial infarction, and the *ACTN3* X allele was associated with a higher risk of developing insulin resistance, which is a known factor of increased mortality [47,48]. Nevertheless, the impact of genetic factors on the higher life expectancy of goalkeepers is still controversial.

The period of systematic training and a professional sports career usually lasts no longer than 25–35 years. Thus, lifestyle and social conditions after a career seem equally important. Factors that may affect the duration of life, which have not yet been mentioned, are as follows: socio-economic status (including professional activity after a sports career), level of education, marital status, access to professional medical care, or level of physical activity after a sports career. The authors were unable to obtain reliable data that would allow us to estimate the aforementioned factors in our players; nevertheless, it is difficult to assume that goalkeepers were somehow different from field players.

It should also be mentioned that height is negatively correlated with life expectancy [49], and goalkeepers are usually taller than other players, but on the other hand, as indicated by the experience of the leading European leagues, goalkeepers tend to end their sporting career at a later age [50]; thus, they remain in the rhythm of intense physical activity for a longer period of time.

Finally, it is worth noting that the data from our analysis strongly demonstrate that elite football players live longer than the general population. Similar results demonstrating longer lifespan were found in a 15-year birth cohort group of elite Polish football players analyzed previously by our group [51] as well as in German, Italian, or Dutch football players investigated by other authors [9,10,11].

### 4.4. Strengths and Limitations

The data from our analysis, which included more than 700 top-class international football players, revealed significant differences in lifespan between goalkeepers and field players. To the best of our knowledge, this is the first study to show a significantly higher lifespan in goalkeepers than in other players. The data are robust and consistent in both the participants of the three World Cup tournaments and elite players of the leading European football clubs. The data related to the death dates of the popular athletes are reliable and were confirmed by other sources.

The main limitation is the retrospective nature of our study and the lack of possibility to analyze several important factors that could influence the longevity of professional athletes. We were not able to compare anthropometric and physiological characteristics of the study participants; however, there are some recently published studies analyzing this problem in detail [13,17,21,24].

We were not able to analyze and compare risk factors, such as smoking or alcohol consumption, as well as other important factors influencing health status following the end of the players’ professional sports careers. These limitations are typical of many other retrospective studies investigating the longevity of professional athletes, and have been described in great detail in the papers of Lemez and Baker [6], Taioli [9], and Kuss et al. [10].

The results of our study are valid for top-class football players born before 1923 who were at the peak of their professional careers during the 1940s and 1950s, which was more than 60 years ago. Present-day professional sports are much more demanding and more commercialized. Although the methods of training and health control of elite football players are better today, one cannot say that the health risks related to professional sports should be neglected.

Despite several methodological limitations, the results of our study are original and intriguing, and could potentially stimulate interest in undertaking further studies that extensively analyze the reasons for lifespan differences in team sports players in relation to their positions on the pitch.

## 5. Conclusions

Goalkeepers have a longer life expectancy than field football players. This phenomenon observed in goalkeepers may be related to the less demanding physical efforts, lower risk of brain trauma caused by repeated head–ball contact, and longer duration of sports career in comparison to players occupying other positions on the pitch. However, these hypotheses require further study.

## Figures and Tables

**Figure 1 ijerph-17-06297-f001:**
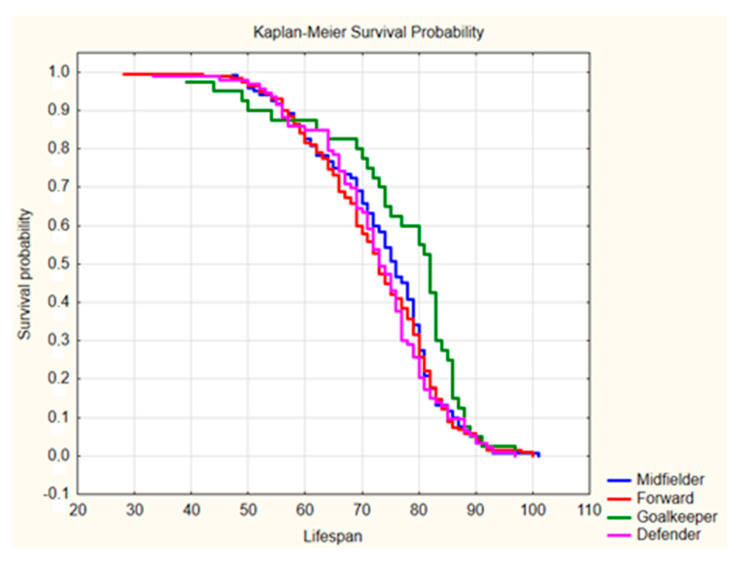
Kaplan–Meier survival probability curves for players who participated in the World Cup tournaments of 1930, 1934, and 1938. Source: self-calculated data based on www.worldfootball.net.

**Figure 2 ijerph-17-06297-f002:**
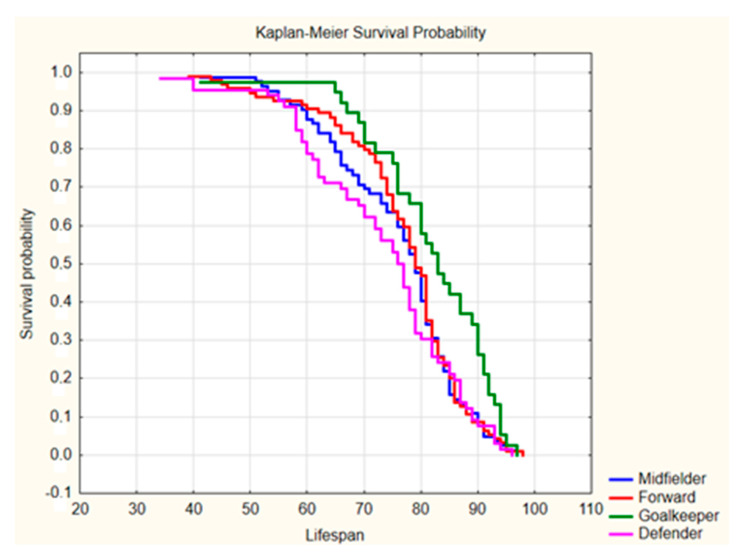
Kaplan–Meier survival probability curves for players playing for leading clubs in the top European leagues during the season 1946/1947. Source: self-calculated data based on www.worldfootball.net.

**Table 1 ijerph-17-06297-t001:** The lifespan of elite football players born before 1922 with regard to the position occupied on the pitch.

Position	World Cup (1930, 1934, 1938)			European Leagues (Season 1946/1947)		
	Median Age (QD)	Survivability up to 85 Years	*N*	Median Age (QD)	Survivability up to 85 Years	*N*
Goalkeeper	82.0 (±7.0)	27.5%	40	83.0 (±7.5)	44.7%	38
Defender	73.0 (±7.0)	12.9%	93	76.5 (±10.5)	24.2%	66
Midfielder	76.0 (±7.8)	13.3%	120	79.0 (±8.5)	22.0%	82
Forward	73.0 (±8.5)	12.1%	190	79.0 (±5.5)	23.4%	94
Total	74.0 (±7.5)	14.0%	443	79.0 (±7.8)	26.1%	280
Total (outfield players)	74.0 (±8.0)	12.7%	403	78.0 (±8.0)	23.1%	242
*p*-value *	0.0047	0.0102		0.0023	0.0048	

* *p*-Value refers to comparison between goalkeepers and field players (analyzed together). Source: self-calculated data based on www.worldfootball.net.

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
