# Peer review of "Goalkeepers Live Longer than Field Players: A Retrospective Cohort Analysis Based on World-Class Football Players"

_ijerph, 2020, doi:10.3390/ijerph17176297_

Round 1
Reviewer 1 Report
The article presented is very original and interesting.
The choice of methodology is very innovative and its results can be of great use for training and sports practice.
The research also dismantles, with evidence, the myth that football goalkeepers live less than other football players. In this sense, the objective and methodology is very original and brilliant.
It is certain that the introduction is very brief and that it would be interesting to deepen more in aspects related to the quality of life, the life expectancy concretely in football players.
The methodology used is original and appropriate. As well as the results and the discussion.
The conclusions are congruent with the work presented.
Author Response
Dear Reviewer,
Thank you very much for your valuable revision. We are glad that our manuscript meets your expectations.
We appreciate your time and effort.
Best regards,
Witold Śmigielski and co-authors.
Reviewer 2 Report
This research manuscript describes the influence of position occupied on the football pitch towards the life duration of the football players. This research was conducted on 723 international football players born up to 1922 who have taken part in either the first three World Cups (Uruguay 1930, Italy 1934 and France 1938) or the 1946/47 season of the main European leagues (e.g., English, Spanish, Italian, German, French, Scottish and Swedish). The authors found that goalkeepers have a longer lifespan compared to the outfield players. Additionally, the survivability of goalkeepers up to 85 years is significantly higher than that of the outfield players. Factors that may contribute to the high lifespan in goalkeepers are training characteristics, intensity of physical activity in game, and risk of head injury. Goalkeepers spend less time at high movement intensities and have lower risk of head injury compared to the outfield players in both training and game. Overall, the manuscript is interesting and well-written. However, there are a few comments that the authors need to address.
1. Lines 57-58: The authors should explain why they opted for players who have taken part in the 1946/47 season of the main European league.
2. Table 1: Add “age” after “Median”.
3. Table 1: Replace “field” with “outfield”.
Author Response
Dear Reviewer,
Thank you very much for your valuable revision.
Regarding point 1.
We have started our analysis from World Class participants (1930, 1934, 1938). After obtaining the results showing that goalkeepers seem to live longer than field players, we wanted to check if these intriguing findings will be confirmed also in other independent group of world-class football players. Taking into consideration the leading position of European teams in world football market, we decided to choose the main European leagues. We have chosen the first seasons after II World War to avoid the need of eliminating (from analysis) players who died during the war.
Regarding point 2.
Done.
Regarding point 3.
Done.
We appreciate your time and effort.
Best regards,
Witold Śmigielski and co-authors.
Reviewer 3 Report
Thank you for the opportunity to review this manuscript. The authors explored whether the life duration of football players depends on the position occupied on the pitch. The authors explored whether the life duration of world-class football players depends on the position occupied on the pitch using a retrospective cohort data from a website www.worldfootball.net. The research has led to the results of high-interest topics with novel ideas.
I have the following concerns.
- The author used the analysis of variance (ANOVA) Kruskal-Wallis test to compare the lifespan of players according to their field position (4 groups). Also, they compared the lifespan between goalkeepers and field players with the Mann-Whitney test. Please explain how you dealt with type-I error inflation in post-hoc analyses (e.g., by Bonferroni corrections). The authors should perform re-analysis and correct the results.
- The authors used the website "www.worldfootball.net" as the main source of information. I couldn't judge the quality of information on this website was valid by the description in the text and the website browsing. Please add additional information.

Author Response
Dear Reviewer,
Thank you very much for your valuable comments.
Regarding point 1.
As you have mentioned, after Kruskal-Wallis analysis of variance (ANOVA), we compared goalkeepers with players on other positions using the Mann-Whitney test and presented the highest
p-value in the table (goalkeeper vs. defender: p=0.0096; goalkeeper vs. midfielder: p=0.0285; goalkeeper vs. forward: p=0.0047). Inspired by your suggestion we decided to use post-hoc analysis for Kruskal-Wallis analysis of variance (ANOVA), i.e. multiple comparison test (Dunn test) in STATISTICA software. The obtained result for Word Cup participants indicated that lifespan of goalkeepers differed significantly in comparison to forwards and defenders (p=0.0234 and p=0.0475, respectively), but differences of lifespan of goalkeepers and midfielders were statistically insignificant (p=0.1959). Nevertheless, it is worth adding that the lifespans of field players did not differ (see: table R1).
Table R1, Parameters of multiple comparison as post-hoc analysis for Kruskal-Wallis ANOVA (World Cup participants)
The obtained result of multiple comparison test for players of the European leagues indicated that lifespan of goalkeepers differed significantly in comparison to defenders and differences of lifespan of goalkeepers and midfielders or forwards were close to the significance level (p=0.0919 and p=0.1079, respectively). Consequently, lifespans of field players did not differ (p=0.7152 or higher; see table R2).
Table R2, Parameters of multiple comparison as post-hoc analysis for Kruskal-Wallis ANOVA (European league players)
Taking into consideration the fact that differences of lifespan of field players are statistically insignificant of World Cup participants and European league players as well, one can analyse them together (i.e. defenders, midfielders and forwards). If so, the difference of lifespan of goalkeepers and field players were statistically significant (Mann-Whitney test, World Cup players: p=0.0047; European League: p=0.0023). Consequently, we decided to compare the surviving rate until 85th birthday also for goalkeepers and field players (analyzed together). So, there is still strong statistics evidence that lifespan of goalkeepers is higher than field players. Information of above mentioned results and analyses were added to the main text of our manuscript (lines 81-83 and 93-105).
Regarding point 2.
Worldfootball platform is a worldwide website collecting information about football players and football teams all over the world. It contains details of over 450 thousands football players from over 30 thousand teams. What is more, selected player’s details were also checked in other sources. Our analysis was based on world-class and well-known football players so main data of their birth or death dates were open to public information.
We appreciate your time and effort.
(tables available in attached files)
Best regards,
Witold Śmigielski and co-authors.

Round 2
Reviewer 3 Report
Thank you for the opportunity to re-review this manuscript. The authors addressed appropriately to all my comments. The manuscript as a whole is that it is well written and clear. The research might be helpful for identifying those who need intensive preventive support.